# Disentangling Content from Style to Overcome Shortcut Learning: A Hybrid Generative-Discriminative Learning Framework

## Abstract

Despite the remarkable success of Self-Supervised Learning (SSL), its generalization is fundamentally hindered by Shortcut Learning, where models exploit superficial features like texture instead of intrinsic structure. We experimentally verify this flaw within the generative paradigm (e.g., MAE) and argue it is a systemic issue also affecting discriminative methods, identifying it as the root cause of their failure on unseen domains. While existing methods often tackle this at a surface level by aligning or separating domain-specific features, they fail to alter the underlying learning mechanism that fosters shortcut dependency. To address this at its core, we propose HyGDL (Hybrid Generative-Discriminative Learning Framework), a hybrid framework that achieves explicit content-style disentanglement. Our approach is guided by the Invariance Pre-training Principle: forcing a model to learn an invariant essence by systematically varying a bias (e.g., style) at the input while keeping the supervision signal constant. HyGDL operates on a single encoder and analytically defines style as the component of a representation that is orthogonal to its style-invariant content, derived via vector projection. This is operationalized through a synergistic design: (1) a self-distillation objective learns a stable, style-invariant content direction; (2) an analytical projection then decomposes the representation into orthogonal content and style vectors; and (3) a style-conditioned reconstruction objective uses these vectors to restore the image, providing end-to-end supervision. Unlike prior methods that rely on implicit heuristics, this principled disentanglement allows HyGDL to learn truly robust representations, demonstrating superior performance on benchmarks designed to diagnose shortcut learning.

## 1 Introduction

Self-Supervised Learning (SSL) has recently emerged as a dominant paradigm in representation learning (Grill et al., 2020; Chen et al., 2020a; Siméoni et al., 2025; Gui et al., 2024; He et al., 2020). Consequently, a significant body of research has aimed to enhance its domain generalization, often by addressing the model's well-documented texture bias (Geirhos et al., 2019). We argue, however, that such approaches often treat the symptom rather than the cause. In this work, we posit that poor generalization stems from a more fundamental problem: the inherent tendency of models towards Shortcut Learning (Geirhos et al., 2020), wherein they exploit superficial features (e.g., texture) that are spuriously correlated with the learning objective, instead of learning the intrinsic, generalizable structure of the data. For clarity in our analysis, we term this well-documented phenomenon the First-Order Shortcut.

Such an inductive bias is not always a flaw, it can be highly informative for specific downstream tasks such as fine-grained classification. For pre-training, however, where the goal is to cultivate general and transferable representations, such a premature commitment to any superficial feature is fundamentally limiting. Herein lies the central paradox of Self-Supervised Learning: the very mechanism that enables it to learn without human labels, the pretext task, is also the primary source of this harmful bias. The implicit objectives of these tasks, from reconstruction in MAE (He et al.,

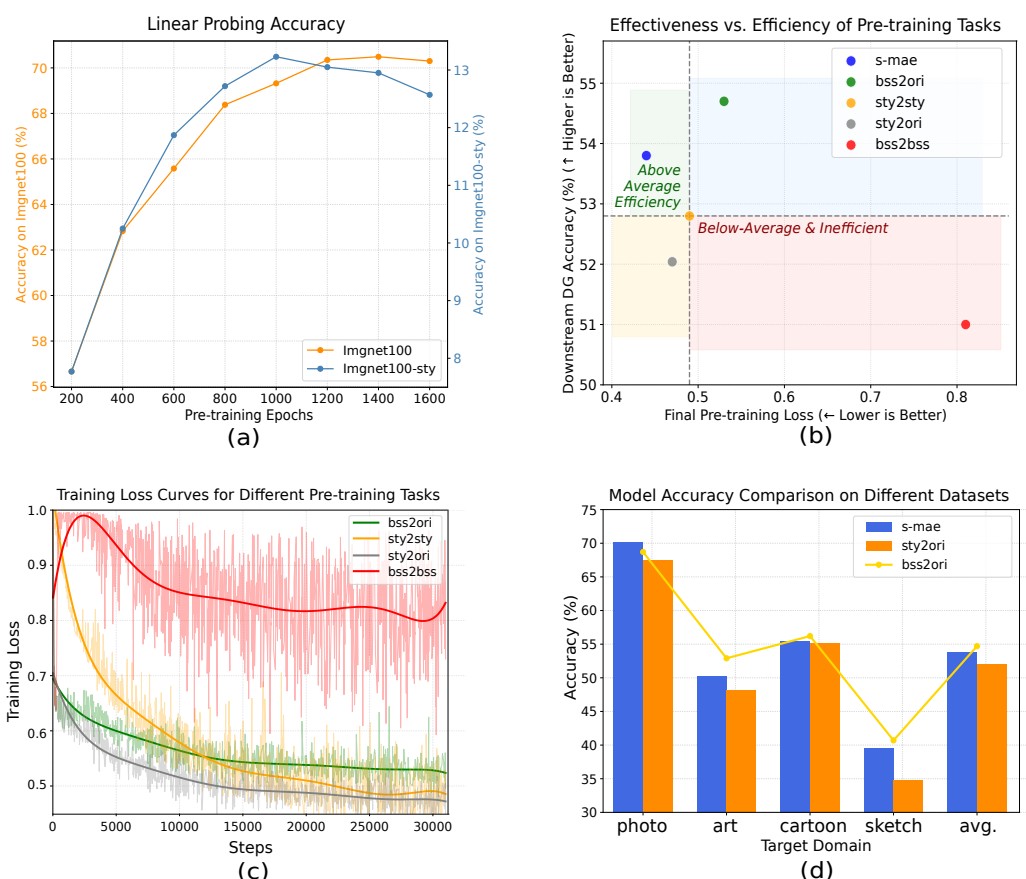

Figure 1: The results of motivation experiments. "s-mae": standard mae with our designed decoder. "bss": an image enhancement method based on Fourier transform (Scalbert et al., 2024). "sty": the Adaptive Instance Normalization (AdaIN) (Huang & Belongie, 2017) method for style transfer. "bss2bss": from enhanced image to reconstruct enhanced image. "bss2ori": from enhanced image to reconstruct original image.

2022) to contrastive alignment in SimCLR and MoCo (Chen et al., 2020a;b; He et al., 2020; Chen et al., 2020c), inevitably create supervisory signals that inadvertently incentivize models to latch onto superficial cues, as they often provide the path of least resistance to minimizing the training objective.

To move beyond indirect evidence for First-Order Shortcut in SSL from prior work and to directly probe our hypothesis, we designed a diagnostic experiment that reveals the dynamics of shortcut learning over time. We established a source domain (standard ImageNet-100) and a stylized target domain (the same images processed with Huang & Belongie (2017)). An MAE model was pre-trained exclusively on the source domain, while its performance on both domains was tracked throughout. (For detailed experimental settings, please refer to Section [3.2] and Appendix)

A striking phenomenon emerged: while the model's in-domain performance consistently improved, its out-of-domain generalization on the stylized data peaked mid-training and then entered a steady decline ((a) in Figure 1). Crucially, this is distinct from standard overfitting, where in-domain validation performance would typically degrade as well. Here, the divergence indicates a more specific pathology: the model begins to overfit to the source domain's shortcuts (i.e., texture), actively trading generalizable structure for domain-specific cues. This reveals that prolonged pre-training does not merely lead to diminishing returns; it actively engineers a representation that is brittle and over-specialized to the source domain's superficial statistics. The model effectively unlearns robust

features in favor of shortcuts. This direct observation demonstrates that the standard pretext task, left unchecked, is not a neutral learner but an active agent of harmful bias, making a fundamental change to its mechanism not just beneficial, but necessary.

A seemingly straightforward solution is to employ style augmentation, but our investigation reveals this to be a pitfall. Naive strategies (e.g., sty2sty) lead to a Second-Order Shortcut (the model bypasses content understanding by simply learning to replicate style textures) or a Reconstruction Dilemma (Figure 1. (b), (c)) demonstrating their unviability. Encouragingly, more sophisticated heuristics like BSS (bss2ori in Figure 1. (d)) avoid this degradation and achieve performance comparable to the MAE baseline. However, the fact that even this better design fails to deliver substantial gains reveals a fundamental limitation, suggesting that implicit, heuristic mechanisms have reached their ceiling.

This impasse motivates our work: to unlock this direction's potential, we propose the Invariance Pre-training Principle and its concrete realization, the HyGDL framework.

**The Invariance Pre-training Principle**: A pre-training task must systematically vary a nuisance bias $B$ (e.g., style, texture) at the model's input, while ensuring the supervisory signal remains invariant to $B$. This design compels the model to learn representations that are robust to the variations of $B$, as relying on any transient instance of the bias will fail to consistently minimize the training objective.

Guided by this principle, we introduce HyGDL, a hybrid generative-discriminative learning framework that serves as its concrete realization. Its design centers on a novel style-conditioned reconstruction task, which provides the ultimate supervision for the entire disentanglement process. Unlike standard masked image modeling, our decoder is tasked to reconstruct the image from two explicitly separated inputs: a style-agnostic content representation and a dedicated style vector.

To produce these disentangled inputs, HyGDL employs a two-stage analytical process. First, a self-distillation objective learns a stable, style-invariant "content direction". Subsequently, an analytical disentanglement step leverages this direction to geometrically decompose an image's representation into its orthogonal content and style components. This synergistic design ensures that the entire pipeline is optimized towards a single, unified goal: learning robust, disentangled representations.

The outcome is a framework that demonstrates strong performance on diagnostic benchmarks, effectively mitigating Shortcut Learning where prior methods failed. More importantly, we see HyGDL not as a final SOTA model, but as a proof-of-concept for a new class of robust SSL frameworks. We believe its principle-guided disentanglement approach offers a promising and generalizable pathway for self-supervision in other challenging domains, from medical imaging to speech recognition.

## 2 METHOD

### 2.1 OVERALL FRAMEWORK

As outlined in the introduction, our HyGDL framework synergizes three key objectives: (1) a style-conditioned reconstruction, (2) a self-distillation objective, and (3) an orthogonality disentanglement step. Figure 2 illustrates the overall architecture. Essentially, the process begins with an input image $A$ and its style-transfer version $A'$, which is generated on-the-fly using the AdaIN method with a pre-trained VGG encoder and a randomly sampled style image. This $(A, A')$ pair is passed through student-teacher encoders, where a self-distillation objective aligns their representations to establish a style-invariant content direction. Using this learned direction, the representation of $A$ and $A'$ is then analytically decomposed into content vector and orthogonal style vectors $A_{style}$ and $A'_{style}$. Finally, the student encoder processes a masked version of $A$ to produce a partial content representation. This partial content, combined with $A_{style}$ and $A'_{style}$, is then fed to the decoder, which is tasked with reconstructing the complete, original image $A$ and style-transfered version $A'$.

To explain the mechanics of how these components work in concert, we will now detail them following the model's computational flow: we first describe how the style-invariant content direction is learned, then how the explicit disentanglement is performed, and finally, how the resulting vectors are utilized in the reconstruction task.

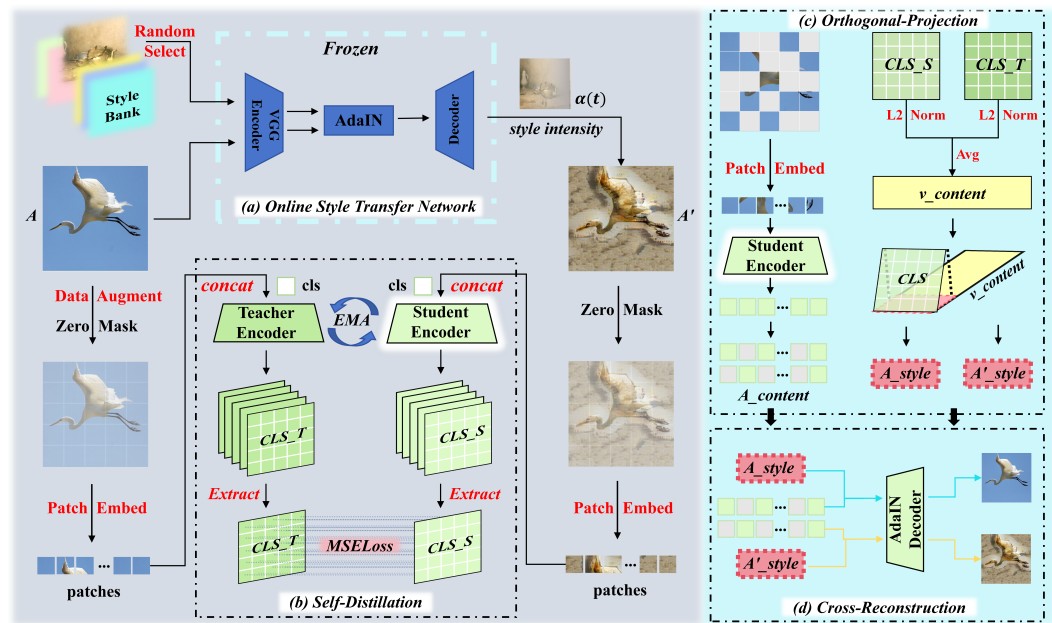

Figure 2: An overview of our HyGDL framework. An input image $A$ and its style-transferred version $A'$ are processed by student-teacher encoders to learn a style-invariant content direction via self-distillation. This direction enables the analytical decomposition of the image's representation into a content vector and a style vector. Finally, a decoder takes the partial content from a masked version of $A$ and the style vectors to reconstruct $A$ and $A'$.

## 2.2 LEARNING THE STYLE-INVARIANT CONTENT DIRECTION

### 2.2.1 CONSTRUCTING POSITIVE PAIRS VIA ONLINE STYLE TRANSFER

A core component of our self-distillation objective is the construction of challenging positive pairs $(A, A')$ that share content but differ drastically in style. To effectively learn style-invariance, the style perturbation must be sufficiently strong and diverse, pushing beyond the simple photometric augmentations used in prior works like BSS.

Furthermore, our Invariance Pre-training Principle mandates that this style variation be performed online and randomly during training. An offline, pre-stylized dataset would present the model with a finite set of styles, risking overfitting to those specific textures rather than learning a generalizable invariance. This requirement rules out computationally intensive methods like GAN-based style transfer (Zhu et al., 2020), which are ill-suited for on-the-fly application within a pre-training loop.

Consequently, we adopt AdaIN as our mechanism for online style transfer. AdaIN's efficiency and effectiveness make it an ideal choice, as it can stylize an image in a single feed-forward pass without requiring a separate, complex generator network. For each image $A$ in a batch, we randomly sample a style image from a diverse style dataset and use AdaIN to transfer its style, producing the stylized view $A'$. Specifically, AdaIN operates by aligning the mean and variance of the content image's feature map with those of the style image's feature map, thereby replacing the content's style statistics with the new ones.

### 2.2.2 IDENTIFYING THE CONTENT DIRECTION VIA SELF-DISTILLATION

Given an image $A$ and its style-transferred version $A'$, the fundamental challenge is to isolate their shared, style-invariant component, the content, from their differing style components. A stable method for identifying this invariant content is the cornerstone of our disentanglement framework.

While a conventional contrastive approach using an InfoNCE loss (Chen et al., 2020a) might seem applicable, we found it suboptimal for the fine-grained semantic alignment required between our

strong positive pairs $(A, A')$. This led us to reframe the problem: the core task is not to distinguish the pair from negatives, but to ensure their representations are identical in the content dimension. This objective aligns perfectly with the philosophy of self-distillation. Inspired by DINO and BYOL, we adopt a student-teacher architecture built upon the MAE encoder. By forcing the student encoder to predict the teacher's output from a stylized view, the model acts as an information bottleneck. It is incentivized to discard the variable, non-essential information (style) and preserve only the shared, invariant information (content).

As shown in part (b) of Figure 2, given an original image $A$ and its style-transferred version $A'$, we treat them as a strong positive pair. The student encoder $f_{\Theta_s}$, which is the primary encoder of our HyGDL model, takes the stylized image $A'$ as input. The teacher encoder $f_{\Theta_t}$ takes a separately augmented view of the original image $A$ as input. To obtain a global representation for the alignment task, we process each image through its respective encoder without applying any mask (i.e., with a blank mask), and use the output of the $[CLS]$ token as the representation. The student's output is denoted as $z_s = f_{\Theta_s}(A')$, and the teacher's output is $z_t = f_{\Theta_t}(A)$.

While methods like DINO employ a cross-entropy loss over softmax-normalized distributions, we found this approach to be suboptimal for our reconstruction-centric framework. A distributional loss, while excellent for learning high-level semantics, can lose some of the fine-grained feature geometry crucial for dense prediction tasks. Therefore, we adopt a direct feature alignment objective using the Mean Squared Error (MSE) loss, similar to BYOL. This encourages a precise vector-to-vector correspondence in the feature space, which is more beneficial for the subsequent reconstruction task.

The loss is computed on L2-normalized embeddings. Let $z_s$ and $z_t$ be the L2-normalized outputs of the student and teacher respectively. The distillation loss is then defined as:

$$L_{\text{distill}} = \|\bar{\mathbf{z}}_s - \bar{\mathbf{z}}_t\|_2^2 \propto 2 - 2 \cdot \frac{\langle \mathbf{z}_s, \mathbf{z}_t \rangle}{\|\mathbf{z}_s\|_2 \cdot \|\mathbf{z}_t\|_2}$$

where $\langle , \rangle$ denotes the inner product. Minimizing this MSE is equivalent to maximizing the cosine similarity between the student and teacher representations.

The teacher's weights $\theta_t$ are an Exponential Moving Average (EMA) of the student's weights $\theta_s$, and the teacher branch operates under a stop-gradient condition. This design compels the student to learn representations that are robustly invariant to style, establishing the stable content direction necessary for the next step of our framework.

## 2.3 EXPLICIT DISENTANGLEMENT VIA ORTHOGONALITY

Our approach to disentanglement is predicated on the manifold hypothesis, which posits that our encoder learns to map high-dimensional images onto a more structured, lower-dimensional feature manifold. Assuming local linearity within this learned manifold, we can employ a principled geometric decomposition. This allows us to move beyond implicit heuristics and analytically separate content from style.

Building on the learned style-invariant content direction, we analytically decompose each representation based on a clear geometric principle: style is the component of a representation that is orthogonal to its content. This explicit, geometric separation moves beyond the implicit heuristics of prior work.

Given an original image $A$ and its stylized version $A'$, we already obtained their respective $[CLS]$ token representations from the encoder, $z_s$ and $z_t$. To ensure the content direction is stable and robust against noise from any single view, we define the content direction vector $v_c$ as the normalized average of the L2-normalized representations from both the student and teacher encoders. This provides a stable estimate of their shared content axis:

$$v_c = L2Normalize(\frac{L2Normalize(z_s) + L2Normalize(z_t)}{2})$$

For simplicity and interpretability, we instantiate content as a single invariant direction. In Appendix, we further extend this decomposition to multi-dimensional subspaces and observe consistent trends. We also include sensitivity analyses of different styles.

With this content direction vector $v_c$ (which is a unit vector), we can decompose any given representation (e.g., $z_s$) into a content component $c_A$ and a style component $s_A$ via vector projection. The content component is the projection of $z_s$ onto $v_c$:

$$c_A = \langle z_s, v_c \rangle \cdot v_c$$

The style component is then derived as the orthogonal residual, capturing all information in the representation that is independent of the content direction:

$$s_A = z_s - c_A$$

This raw style vector $s_A$, while geometrically pure, represents a direction of stylistic deviation in the feature space and may not be in an optimal format for direct use by the decoder. Therefore, we introduce a learnable style projector $g_s(\cdot)$ to bridge this gap. This projector, implemented as a small multi-layer perceptron (MLP), is tasked with mapping the raw geometric style vector into a more structured and functionally effective style embedding. The final style vector is:

$$style_{vector} = g_s(s_A)$$

We present the core principle using a single content direction for clarity; in practice, our main experiments use a two-dimensional content subspace ($k = 2$) which yields the best trade-off (see Appendix A.5, Table 11).

### 2.4 STYLE-CONDITIONED RECONSTRUCTION

The final component of our framework is the style-conditioned reconstruction task, which serves as the ultimate supervisor for the entire disentanglement process, ensuring all modules work synergistically towards a unified goal. This task requires a decoder capable of synthesizing an image from two disentangled inputs: a style-agnostic content representation and a dedicated style vector.

To adapt the standard MAE decoder for two inputs, we insert multiple AdaIN layers at various depths in the decoder, allowing for the progressive infusion of style information into the content features across different levels of the reconstruction process. This design creating a deliberate methodological symmetry, as we use AdaIN both to generate stylized training data and to later fuse the disentangled style vector during reconstruction.

#### 2.4.1 SELF-RECONSTRUCTION FOR FIDELITY

The primary reconstruction objective is to restore the original, non-stylized image $A$ from its own disentangled components. This task anchors the learning process and ensures that the separated vectors retain sufficient information for faithful synthesis. The decoder $D$ takes the partial content representation from the masked image, $f_{\Theta_s}(A_{masked})$, and the corresponding refined style vector, $style_{vector}$, as inputs. The self-reconstruction loss $L_{self}$ is the MSE computed on the set of masked patches $M$:

$$L_{self} = \frac{1}{|M|} \sum_{i \in M} \|D(f_{\Theta_s}(A_{masked}), style_{vector}) - A_i\|_2^2$$

#### 2.4.2 CROSS-RECONSTRUCTION FOR ROBUSTNESS

To further enforce that the content representation is truly style-agnostic, we introduce a cross-reconstruction objective. This task acts as a powerful regularizer by challenging the decoder to synthesize a new image from content and style derived from different sources. Specifically, given the content from image $f_{\Theta_s}(A_{masked})$ and the style from $A'$, the decoder must reconstruct $A'$. The loss is formulated as:

$$L_{cross} = \frac{1}{|M|} \sum_{i \in M} \|D(f_{\Theta_s}(A_{masked}), style'_{vector}) - A'_i\|_2^2$$

This objective penalizes any residual style information in the content vector, as it would conflict with the target style and lead to a high reconstruction error.

The gradient flow from this seemingly simple loss function provides a rich and dense supervisory signal that holistically integrates and optimizes the entire HyGDL framework. In an end-to-end fashion, it jointly imposes fine-grained constraints on three core components. For the encoder $f_{\Theta_s}$, it forces the encoder not only to learn high-quality local features sufficient for reconstruction but also to produce a global representation that is amenable to orthogonal decomposition. For the style projector $g_s(\cdot)$, it drives the projector to transform the geometric style residual vector into a functionally effective style embedding for the decoder. For the decoder $D$, it trains the decoder to become a fusion expert, learning to efficiently combine pure content information and a separate style embedding to synthesize a high-fidelity image.

## 2.5 Training with a Progressive Three-Stage Curriculum

Our final training objective is a weighted sum of three distinct losses. A key challenge is to balance these objectives to ensure stable training and avoid failure modes like the Second-Order Shortcut. We address this by designing a progressive three-stage curriculum that schedules not only the loss weights but also the task difficulty itself. The style transfer mechanism includes a scheduled stylization coefficient $\alpha(t)_{style} \in [0, 0.5]$, which controls the interpolation between the original and stylized feature statistics. The total loss is:

$$L_{total} = L_{self} + \lambda_{align}(t)L_{distill} + \lambda_{cross}L_{cross}$$

where $\lambda_{align}(t)$ is a scheduled weight. The curriculum unfolds as follows:

1. Foundational Representation Learning. In this initial stage, we set both $\alpha(t)_{style}$ and $\lambda_{align}(t)$ to 0. The framework focuses on learning basic augmentation invariance via self-distillation and learning the reconstruction task.

2. Style Invariance Warm-up. We gradually ramp up $\alpha(t)_{style}$ and $\lambda_{align}(t)$ from 0 to max. This progressively increases the style difference between the student's and teacher's inputs, forcing the encoder to learn a representation that is truly invariant to drastic style perturbations.

3. Full Synergistic Training. In this final stage, the model, now capable of robustly disentangling content, is challenged with the full cross-synthesis task, ensuring all components are synergistically optimized.

## 3 Experiments

### 3.1 Datasets

To better evaluate our model, we conduct a series of diagnostic, comparative, and ablation studies on datasets including ImageNet-100, PACS and DomainNet (Peng et al., 2019). Appendix details the settings and training overhead: HyGDL requires 40% more GPU-hours and higher peak memory than MAE (Table 12), though its inference cost is identical as it only uses the student encoder.

ImageNet-100: a 100-class subset of the full ImageNet dataset. PACS: a standard domain generalization benchmark consisting of 7 object classes across 4 visually distinct domains (Photo, Art, Cartoon, and Sketch). DomainNet: a large-scale domain generalization benchmark consisting of 345 object classes across 6 visually distinct domains (Clipart, Infograph, Painting, Quickdraw, Real, and Sketch).

### 3.2 Diagnostic Experiments for First-Order Shortcut

To empirically diagnose the dynamics of the First-Order Shortcut in a standard SSL paradigm, we conduct a controlled experiment using the MAE framework. We pre-train a standard MAE model on the ImageNet-100 (IN-100) source domain for 1600 epochs, saving model checkpoints every 200 epochs. At each checkpoint, we evaluate the quality of the learned representations by performing linear probing on two target domains: the original in-domain test set (IN-100) and an out-of-domain, stylized test set (IN-100-sty). We chose linear probing over full fine-tuning as it provides a more direct and frozen assessment of the encoder's representation quality, minimizing confounding effects from adapting the entire model.

Table 1: Results of linear probing on models pre-trained using the MAE method. All reported accuracies are in percent (%).

| | Linear Probing on IN-100 Pre-trained | | | | Linear Probing on IN-100-sty Pre-trained | | | |
| | Test on IN-100 | | Test on IN-100-style | | Test on IN-100 | | Test on IN-100-style | |
| Epochs | Top-1 | Top-5 | Top-1 | Top-5 | Top-1 | Top-5 | Top-1 | Top-5 |
|---|---|---|---|---|---|---|---|---|
| 200 | 56.66 | 83.13 | 7.77 | 22.62 | 31.95 | 62.47 | 20.97 | 46.38 |
| 400 | 62.83 | 86.85 | 10.25 | 26.53 | 38.95 | 68.97 | 24.98 | 51.13 |
| 600 | 65.58 | 88.61 | 11.87 | 29.02 | 41.50 | 71.00 | 26.58 | 52.65 |
| 800 | 68.38 | 89.30 | 12.72 | 30.00 | 45.36 | 73.01 | 27.61 | 54.34 |
| 1000 | 69.32 | 89.51 | 13.23 | 30.21 | 45.89 | 73.52 | 27.44 | 54.61 |
| 1200 | 70.35 | 89.58 | 13.05 | 30.29 | 45.82 | 73.21 | 27.46 | 53.32 |
| 1400 | 70.49 | 89.53 | 12.95 | 30.09 | 45.47 | 73.11 | 27.25 | 53.31 |
| 1600 | 70.30 | 89.26 | 12.57 | 28.81 | 44.65 | 72.26 | 26.21 | 52.24 |

Table 2: Unsupervised Domain Generalization on PACS and DomainNet subset. Best methods are highlighted in **bold**, and all reported accuracies are in percent (%).

| | PACS | | | | | DomainNet | | | |
| | Target Domain | | | | | Target Domain | | | |
| Method | photo | art | cartoon | sketch | avg. | clipart | infograph | quickdraw | avg. |
|---|---|---|---|---|---|---|---|---|---|
| ERM | 43.29 | 24.27 | 32.62 | 20.84 | 30.26 | 15.10 | 9.39 | 7.11 | 10.53 |
| MoCo V2 | 59.86 | 28.58 | 48.89 | 34.79 | 43.03 | 32.46 | 18.54 | 8.05 | 19.68 |
| SimCLR V2 | 67.45 | 43.60 | 54.48 | 34.73 | 50.06 | 37.11 | 19.87 | 12.33 | 23.10 |
| BYOL | 41.42 | 23.73 | 30.02 | 18.78 | 28.49 | 14.55 | 8.71 | 5.95 | 9.74 |
| AdCo | 58.59 | 29.81 | 50.19 | 30.45 | 42.26 | 32.25 | 17.96 | 11.56 | 20.59 |
| DARLING | 68.86 | 41.53 | 56.89 | 37.51 | 51.20 | 35.15 | 20.88 | 15.69 | 23.91 |
| DIMAE$^+$ | 78.99 | 63.23 | 59.44 | **55.89** | 64.39 | 70.78 | 38.06 | 27.39 | 45.41 |
| BrAD$^+$ | – | – | – | – | – | 68.27 | 26.60 | **34.03** | 42.97 |
| CycleMAE$^+$ | **90.72** | **75.34** | **69.33** | 50.24 | **71.41** | **74.87** | **38.42** | 28.32 | **47.20** |
| MAE w/bss | 66.40 | 47.11 | 54.94 | 35.58 | 51.01 | 46.32 | 20.91 | 9.42 | 25.55 |
| MAE w/sty | 70.59 | 46.58 | 53.84 | 39.91 | 52.73 | 41.92 | 17.53 | 8.41 | 22.62 |
| HyGDL($k = 2$) | 77.31 | 55.37 | 59.13 | 40.72 | 58.13 | 51.40 | 23.81 | 12.54 | 29.25 |

$^+$: Use Imagenet transfer learning.

MAE w/bss and MAE w/sty: Added the bss module and the AdaIN style transfer module on top of the standard MAE, respectively.

The results, presented in Table 1, reveal two critical phenomena. First, a stark performance gap exists between in-domain and cross-domain evaluation. The model pre-trained for 1600 epochs achieves 70.30 % Top-1 accuracy on IN-100 but plummets to just 12.57% on IN-100-sty. This confirms that the learned representation is brittle and heavily reliant on source-domain style cues. Second, and more importantly, the cross-domain performance on IN-100-sty exhibits the characteristic "rise-and-fall" pattern we hypothesized. The accuracy peaks at 13.23% around 1000 epochs and subsequently declines. This provides direct evidence of the First-Order Shortcut: as pre-training continues, the model shifts from learning generalizable features to overfitting on source-domain shortcuts, actively harming its generalization.

## 3.3 COMPARISON WITH STATE-OF-THE-ART

In the following experiments, we compare our framework against representative SSL and Unsupervised Domain Generalization (UDG) methods. To maintain focus, this section presents the main results on two key benchmarks. For a more comprehensive evaluation, including results on additional datasets, please refer to the Appendix, and there we also observe an interesting asymmetric generalization phenomenon, which is closely related to the characteristics of the source domains.

**Analysis of Results.** As shown in Table 2, our HyGDL framework demonstrates strong performance on both PACS (58.13%) and the DomainNet subset (29.25%).(1) **Superiority over SSL Baselines:** HyGDL significantly surpasses standard SSL methods like SimCLR V2 and MAE, confirming the effectiveness of our explicit disentanglement approach for OOD robustness. (2) **Validation of Principled Design:** It also substantially outperforms heuristic-based approaches (MAE

Table 3: Ablation studies on PACS. Best methods are highlighted in **bold**, and all reported accuracies are in percent (%).

| | PACS | | | | |
|---|---|---|---|---|---|
| | Target Domain | | | | |
| **Method** | **photo** | **art** | **cartoon** | **sketch** | **avg.** |
| no-distill | 67.30 | 48.73 | 56.74 | 34.3 | 51.76 |
| no-recon | 25.86 | 42.48 | 42.44 | 39.98 | 37.69 |
| no-cross-recon | 74.85 | 52.58 | **59.17** | 38.71 | 56.33 |
| CLI | 69.88 | 48.24 | 57.21 | 34.49 | 52.45 |
| SDD | 59.40 | 38.43 | 47.27 | 25.53 | 43.40 |
| FiLM | 60.11 | 46.78 | 54.19 | 35.07 | 49.04 |
| LN | 69.88 | 48.24 | 57.21 | 34.49 | 52.46 |
| HyGDL | **77.31** | **55.37** | 59.13 | **40.72** | **58.13** |

w/bss and MAE w/sty), validating our hypothesis that a principled mechanism is superior. (3) **Competitive SOTA Performance:** Crucially, while methods like CycleMAE leverage large-scale ImageNet pre-training, our HyGDL is pre-trained from scratch on only the source domain. Despite this significant disadvantage, it still achieves highly competitive results, outperforming many specialized UDG methods like DARLING. This highlights the exceptional data efficiency and powerful generalization capability learned by our approach.

### 3.4 ABLATION STUDIES

To validate the effectiveness of our key design choices, we conduct a series of comprehensive ablation studies on the PACS dataset. We start with our full HyGDL model as the benchmark and analyze the impact of removing or altering its core components. The results are summarized in Table 3. And the findings confirm the importance of each component in our HyGDL framework.

First, the results confirm the necessity of our core objectives. Removing the self-distillation (no-distill), self-reconstruction (no-recon), or cross-reconstruction (no-cross-recon) loss each leads to significant performance degradation, with the self-distillation objective being the most critical. Second, we validate our choice of alignment strategy. Our MSE-based self-distillation significantly outperforms both Contrastive Learning (CLI) and DINO's cross-entropy loss (SDD), validating our hypothesis that a direct feature-space alignment is better suited for our reconstruction-centric framework. Finally, our architectural choices are proven to be effective; replacing the AdaIN-based decoder with alternatives like LayerNorm (LN) or FiLM (Perez, 2019) results in a noticeable performance drop, demonstrating the superiority of our style injection mechanism.

### 4 CONCLUSION

In this work, we addressed the fundamental problem of shortcut learning in SSL by proposing the Invariance Pre-training Principle and its concrete realization, the HyGDL framework. Our approach introduces a novel, explicit disentanglement mechanism that synergizes a self-distillation objective to identify a style-invariant content direction with a style-conditioned reconstruction task. This design allows the framework to analytically separate style from content and provides robust, end-to-end supervision for the learned representations.

Our experiments demonstrate that HyGDL effectively mitigates shortcut learning, outperforming standard SSL and heuristic baselines on multiple UDG benchmarks. Beyond proposing a model, this work provides a proof-of-concept for a principled framework based on explicit geometric disentanglement. Nonetheless, certain limitations remain: performance degrades in highly sparse domains such as Quickdraw, and results are sensitive to the choice of style-source, richer/style-diverse sources (e.g., Art) provide a stronger training signal (see Appendix A.4.1 and A.4.3). Future work may extend this principle to other modalities, including audio and medical imaging, and explore strategies to reduce sensitivity to domain characteristics. We hope this study encourages further research toward robust, interpretable, and generalizable representation learning.

## 5 REPRODUCIBILITY STATEMENT

To ensure the reproducibility of our findings, we provide the complete source code, pre-training scripts, and fine-tuning configurations in an anonymous Git repository here: `https://anonymous.4open.science/r/HyGDL-45E1`. All experimental details, including dataset processing, hyperparameters, and the specific three-stage training curriculum, are thoroughly documented in Section 3 and the Appendix.

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

## A APPENDIX

### A.1 A STATEMENT ON LARGE LANGUAGE MODELS USAGE

A Large Language Model (LLM) was utilized as an assistant for manuscript preparation. Its contributions include refining the language and structure of the paper, as well as providing suggestions for articulating the theoretical underpinnings of our method. All core research ideas and experiments were conducted by the human authors.

### A.2 RELATED WORKS

#### A.2.1 SELF-SUPERVISED LEARNING PARADIGMS

SSL aims to learn meaningful representations from large-scale unlabeled data by leveraging carefully designed pretext tasks, thereby obviating the need for manual annotation. In the field of computer vision, mainstream SSL paradigms can be broadly categorized(Gui et al., 2024; Liu et al., 2023a). One major family is discriminative learning, which trains models to distinguish between similar and dissimilar sample pairs. Seminal works like SimCLR pioneered the contrastive approach (Zhao et al., 2021; Zheng et al., 2021; Xie et al., 2021), relying on numerous negative samples to prevent collapse. To mitigate this dependency, distillation-based methods like BYOL and DINO emerged (Siméoni et al., 2025; Grill et al., 2020; Caron et al., 2020), using a student-teacher architecture with mechanisms like stop-gradient to avoid trivial solutions. Despite their different collapse-prevention strategies, all these methods fundamentally rely on a shared mechanism: constructing learning signals from standard data augmentations (e.g., random cropping, color jitter). Consequently, the invariance they cultivate is inherently limited to "Intra-Domain Invariance", the robustness to low-level photometric and geometric perturbations within a single domain.

Diverging from the discriminative approaches, the generative paradigm, particularly Masked Image Modeling (MIM), draws inspiration from the success of BERT in natural language processing (Zhou et al., 2022; Xie et al., 2022; Devlin et al., 2019; Bao et al., 2022). Leading methods like MAE employ an encoder-decoder architecture, where the model is tasked with reconstructing original pixels from a heavily corrupted input, in which a large portion of image patches are masked. This challenging in-painting task compels the encoder to learn a holistic understanding of object structure and contextual relationships. However, it implicitly incentivizes the encoder to also capture superficial textures (First-Order Shortcut), as these low-level cues are vital for the decoder to perform accurate reconstruction.

Recently, a significant trend has emerged towards hybridizing generative and discriminative (ID/CL) objectives to learn more comprehensive representations. For instance, RePre (Wang et al., 2022) augments contrastive frameworks with a parallel reconstruction branch to enhance local features. CMAE (Huang et al., 2024) designs an auxiliary feature decoder to complement masked views for effective contrastive alignment. Further, SiameseIM (Tao et al., 2023) proposes to reconstruct dense features of one view from another masked view. However, the primary motivation of these works is largely to synergize complementary features (e.g., local details from MIM and global semantics from ID). They rely on the combination of tasks to implicitly encourage disentanglement, but unlike our work, they do not start from the root cause of shortcut learning nor introduce an explicit, geometric constraint (i.e., orthogonality) to fundamentally alter the learning mechanism.

Despite their methodological diversity and empirical success, these paradigms all share a fundamental vulnerability when faced with domain shifts: a susceptibility to Shortcut Learning, which undermines their generalization capabilities. We will dissect this challenge in the following section.

#### A.2.2 SHORTCUT LEARNING: FROM TEXTURE BIAS TO A FUNDAMENTAL CHALLENGE IN SSL

The texture bias of neural networks is one of the most well-documented manifestations of shortcut learning, a phenomenon we term the First-Order Shortcut. The seminal work of Geirhos et al. (2019) first systematically demonstrated that ImageNet-trained CNNs are strongly biased towards texture rather than shape. By constructing a stylized dataset (Stylized-ImageNet), they showed that this texture bias harms model robustness and proposed stylized data augmentation as a remedy

to encourage shape-based representations. In a follow-up work, Geirhos et al. (2020) framed this phenomenon as a specific instance of the broader problem of Shortcut Learning, where models exploit superficial cues that are spuriously correlated with the objective, instead of learning the causal, generalizable rules intended by researchers.

Given that SSL methods often employ the same backbone architectures as supervised learning, a natural question arises: do they inherit the same tendency for Shortcut Learning? Recent work provides preliminary evidence. For instance, PIXMIM (Liu et al., 2023b) improves MAE performance by low-frequency target generation, indirectly suggesting that the standard MAE may over-rely on low-level textures, contrastive learning forces the model to learn Intra-Domain Invariance by strong data augmentation (Scalbert et al., 2021; Kang et al., 2022; Xu et al., 2021), which can be seen as an implicit mechanism to break the simple texture-based dependencies characteristic of the First-Order Shortcut. However, while such works propose targeted solutions, they often lack a systematic diagnosis of the dynamics of shortcut learning in SSL. How, and at what stage of training, do these models begin to overfit to shortcuts? This fundamental question remains largely unanswered and serves as the primary motivation for our study.

### A.2.3 DISENTANGLEMENT FOR ROBUST REPRESENTATION LEARNING

To overcome the limitations of "In-Domain Invariance", a key research direction is the pursuit of a more powerful "Inter-Domain Invariance", realized through domain-invariant representations. The core mechanism to achieve this is feature disentanglement: separating the universal, task-relevant "content" information within a representation from the variable, task-irrelevant "style" or "domain" information.

Current work towards "Inter-Domain Invariance" largely follows two main paths. One line of work relies on multi-component architectures, where methods like DIMAE and CycleMAE (Yang et al., 2022; 2023) use domain-specific decoders to isolate domain information. However, this approach suffers from poor scalability and often requires domain labels, contradicting the ethos of self-supervision. More critically, its generative mechanism can introduce the Second-Order Shortcut we identify, where the model learns to replicate uniform style textures instead of content. The other path pursues implicit disentanglement within a single model. A representative example, BSS (Scalbert et al., 2024), cleverly separates style by swapping amplitude spectra in the Fourier domain. While this heuristic is effective, its success is rooted in the specific properties of signal processing and does not constitute a principled, explicit disentanglement within the deep feature space itself. And unlike approaches that rely on strong architectural priors to enforce a specific, 2D spatial notion of content, operate as post-processing modules on fixed, pre-trained features, or achieve implicit disentanglement through complex, invertible transformations (Wu et al., 2019; Ma et al., 2025; Ngweta et al., 2023), HyGDL is the first to learn an explicit, analytical disentanglement via end-to-end pre-training within a universal single-encoder. It uniquely combines self-distillation (to identify a style-invariant subspace) with geometric orthogonality as the core separation mechanism, all supervised by a style-conditioned reconstruction task, thus forging a novel methodological path.

In summary, prior works have largely focused on designing remedial measures to counteract the symptoms of the First-Order Shortcut. And even sophisticated approaches like multi-component architectures can inadvertently introduce new failure modes, such as the generative pitfall of the Second-Order Shortcut, rather than fundamentally altering the learning mechanism. We argue that a more fundamental solution requires a clear guiding principle to reshape the pretext task itself. To this end, we propose the Invariance Pre-training Principle and introduce our HyGDL framework as its concrete realization. By employing explicit geometric disentanglement, HyGDL directly addresses the learning mechanism to foster the acquisition of style-invariant content.

### A.3 EXPERIMENTS SETTING

### A.3.1 DIAGNOSTIC EXPERIMENT

For the diagnostic experiment, we follow the work of Geirhos et al. (2019) to create the IN-100-sty dataset based on ImageNet-100. The style transfer is performed using the AdaIN method with a pre-trained VGG-19. All training and evaluation are conducted on 1 NVIDIA RTX 4090 GPU. And the training settings are shown in Table 4:

Table 4: The settings of diagnostic experiment.

| Hyperparameters | Pre-training | Linear probing |
|---|---|---|
| backbone | vit-base | – |
| mask ratio | 75% | – |
| batch size | 256 | 2048 |
| epochs | 1600 | 90 |
| optimizer | adamw | sgd |
| adamw betas | (0.9,0.999) | – |
| momentum | – | 0.9 |
| learning rate | 1.5e-4 | 5e-3 |
| learning rate schedule | linear warmp-up + cosine decay | linear warmp-up + cosine decay |
| weight decay | 0.05 | 0 |
| warm-up epochs | 40 | 10 |

Table 5: The settings of comparison experiment.

| Hyperparameters | Pre-training | Fine-tuning |
|---|---|---|
| backbone | vit-small | – |
| mask ratio | 75% | – |
| batch size | 128 | 256 |
| epochs | 1000 | 150 |
| optimizer | adamw | adamw |
| adamw betas | (0.9,0.95) | (0.9,0.95) |
| learning rate | 3e-6 | 5e-4 |
| learning rate schedule | linear warmp-up + cosine decay | linear warmp-up + cosine decay |
| weight decay | 0.05 | 0.05 |
| warm-up epochs | 30 | 10 |
| data augmentation | ResizedCrop + HorizontalFlip | ResizedCrop + HorizontalFlip |
| three-stage curriculum | 0-10, 10-100, 100-1000 epochs | – |
| $\alpha(t)_{style}$ | [0,0.5] | – |
| $\lambda_{align}(t)$ | [0,0.15] | – |
| $\lambda_{cross}$ | 0.1 | – |

Table 6: Asymmetric generalization experiments on DomainNet subset. Best methods are highlighted in **bold**, and all reported accuracies are in percent (%).

| Method | $clipart \cup infograph \cup quickdraw$ | | | | $painting \cup real \cup sketch$ | | | |
|---|---|---|---|---|---|---|---|---|
| | Target Domain | | | | Target Domain | | | |
| | painting | real | sketch | avg. | clipart | infograph | quickdraw | avg. |
| ERM | 9.90 | 9.19 | 5.12 | 8.07 | 15.10 | 9.39 | 7.11 | 10.53 |
| MoCo V2 | 25.35 | 29.91 | 23.71 | 26.32 | 32.46 | 18.54 | 8.05 | 19.68 |
| SimCLR V2 | 24.01 | 30.17 | 31.58 | 28.59 | 37.11 | 19.87 | 12.33 | 23.10 |
| BYOL | 9.50 | 10.38 | 4.45 | 8.11 | 14.55 | 8.71 | 5.95 | 9.74 |
| AdCo | 23.35 | 29.98 | 27.57 | 26.97 | 32.25 | 17.96 | 11.56 | 20.59 |
| DARLING | 25.90 | 33.29 | 30.77 | 29.99 | 35.15 | 20.88 | 15.69 | 23.91 |
| DIMAE$^+$ | 50.73 | 64.89 | 55.41 | 57.01 | 70.78 | 38.06 | 27.39 | 45.41 |
| BrAD$^+$ | 31.08 | 38.48 | 48.71 | 39.24 | 68.27 | 26.60 | **34.03** | 42.97 |
| CycleMAE$^+$ | **52.81** | **67.13** | **56.37** | **58.77** | **74.87** | **38.42** | 28.32 | **47.20** |
| MAE w/bss | 18.38 | 23.32 | 24.6 | 22.1 | 46.32 | 20.91 | 9.42 | 25.55 |
| MAE w/sty | 19.45 | 23.7 | 22.30 | 21.81 | 41.92 | 17.53 | 8.41 | 22.62 |
| HyGDL($k = 2$) | 21.71 | 28.11 | 28.60 | 26.14 | 51.40 | 23.81 | 12.54 | 29.25 |

$^+$: Use Imagenet transfer learning.
MAE w/bss and MAE w/sty: Added the bss module and the AdaIN style transfer module on top of the standard MAE, respectively.

### A.3.2 COMPARISON EXPERIMENT

We compare our framework with representative SSL and UDG methods under the settings of Scalbert et al. (2024) and Peng et al. (2019). Specifically, 100% of the data is used for fine-tuning on PACS and 10% on DomainNet. All experiments are run on two NVIDIA RTX 4090 GPUs, with training details provided in Table 5.

## A.4 MORE EXPERIMENTS

### A.4.1 ASYMMETRIC GENERALIZATION AND SOURCE DOMAIN CHALLENGES IN GENERATIVE PRE-TRAINING

To deeply investigate the behavior of our generative-based hybrid framework on complex, multi-domain benchmarks, we conducted two symmetric transfer experiments on a subset of DomainNet. Setting A (Schematic-to-Naturalistic): We used Clipart, Infograph, Quickdraw as the multi-source training domains and evaluated on Real, Painting, Sketch. Setting B (Naturalistic-to-Schematic): We used Real, Painting, Sketch as the source domains and evaluated on Clipart, Infograph, Quickdraw.

The results, presented in Table 6 reveal an asymmetry in generalization performance. In Setting A, both our HyGDL and the baseline MAE failed to generalize effectively. Conversely, in Setting B, our HyGDL demonstrated a clear advantage, successfully transferring knowledge to the schematic domains. This asymmetry points to a fundamental limitation of MIM when the source domain is visually impoverished. Our investigation suggests the "Quickdraw" domain is the primary cause of the failure in Setting A. Generative-based pre-training relies on reconstructing rich visual context from partial inputs. When the source data is dominated by sparse, schematic line drawings like "Quickdraw", the model is deprived of the necessary low-level texture and contextual cues (as shown in Figure 3). It fails to learn a rich enough feature vocabulary, leading to a catastrophic collapse in performance when transferring to visually dense domains like "Real" or "Painting".

This finding is further supported by the universally low performance of all competing methods when "Quickdraw" is the target domain (as seen in Table 6), confirming its unique and challenging characteristics. This leads us to question the role of "Quickdraw" in standard UDG benchmarks, particularly for evaluating generative-based pre-training methods. While it serves as a difficult test case, its extreme nature may not reflect a realistic domain shift. Its properties make it a pathological source domain for MIM-based approaches, and its suitability as a universal benchmark for all types of UDG methods warrants further discussion within the community.

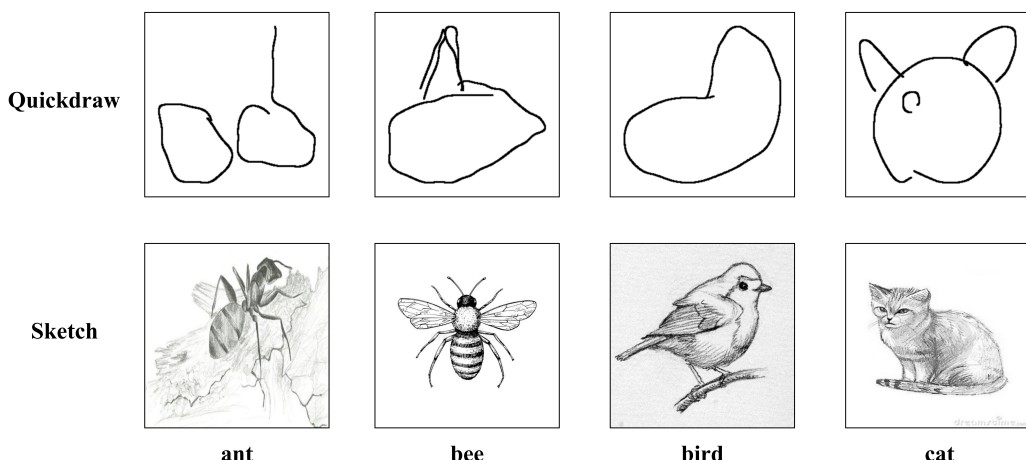

Figure 3: The comparison of four object classes (ant, bee, bird and cat) in subset "Quickdraw" and "Sketch".

### A.4.2 Ablation Studies

In this section, we provide additional ablation studies to further analyze the key components and hyperparameter sensitivity of our HyGDL framework. For these studies, we use our base model configuration with a content subspace dimensionality of $k = 1$ as the benchmark, and all experiments are conducted on the PACS dataset.

**Analysis of Teacher Input Augmentation.** A key mechanism in distillation-based SSL is the asymmetry between the student and teacher inputs, which is crucial for preventing model collapse. To validate its importance in our framework, we compare our standard setup, where the teacher receives a randomly augmented view of the original image, against a variant where the teacher receives the clean, un-augmented original image. As shown in Table 7, removing the teacher-side augmentation leads to a noticeable drop in performance. This confirms that the input asymmetry is vital. It prevents the student from finding a trivial solution (e.g., identity mapping) and compels it to learn a more abstract and robust representation of content by aligning two different views of the same underlying image.

**Sensitivity to Loss Weights.** We analyze the sensitivity of HyGDL to the weights of the self-distillation loss ($\lambda_{distill}$) and the cross-reconstruction loss ($\lambda_{recon}$). We vary the weight of one objective while keeping the others fixed at their default values. Table 8 summarizes the results. For the self-distillation loss, performance peaks around a weight of [0.15], which we use in our main experiments. For the cross-reconstruction loss, the best performance is achieved at [0.1]. Crucially, the results show that HyGDL's performance is relatively stable across a reasonable range of weights for both objectives, indicating that our framework is not overly sensitive to these hyperparameters.

**Impact of Gradient Flow from Cross-Reconstruction.** A cornerstone of our framework is the role of the reconstruction loss as an end-to-end supervisor for the entire disentanglement pipeline. To explicitly validate this, we conduct an experiment to analyze the impact of the gradient flow from the cross-reconstruction loss to the encoder. We compare our full model against a variant where the gradient from $L_{cross}$ is stopped before the encoder (stop gradient), meaning it only trains the style projector and the decoder. The results in Table 9 show a significant performance degradation when the gradient to the encoder is cut off. This experiment provides powerful evidence for our central claim: the cross-reconstruction task is not merely a decoder-training objective but a crucial regularizer for the encoder's representation. The supervisory signal from $L_{cross}$ forces the encoder to produce a "purer," more effectively disentangled content representation that is functionally optimized for the downstream synthesis task. Without this signal, the encoder's representation quality for OOD generalization diminishes.

Table 7: Ablation studies of input augmentation on PACS. Best methods are highlighted in **bold**, and all reported accuracies are in percent (%).

| | **PACS** | | | | |
| | | Target Domain | | | |
| **Method** | **photo** | **art** | **cartoon** | **sketch** | **avg.** |
| un-augmented | 59.88 | 42.23 | 46.2 | 35.89 | 46.05 |
| augmented | **76.17** | **54.0** | **59.51** | **37.82** | **56.88** |

Table 8: Ablation studies of Loss Weights on PACS. Best methods are highlighted in **bold**, and all reported accuracies are in percent (%).

| | **PACS** | | | | |
| | | Target Domain | | | |
| **Loss Weights** | **photo** | **art** | **cartoon** | **sketch** | **avg.** |
| $\lambda_{distill} = 0.1$ | 74.85 | 52.58 | 59.17 | **38.71** | 56.33 |
| $\lambda_{distill} = 0.15$ | 74.97 | 53.02 | **61.56** | 37.92 | 56.87 |
| $\lambda_{distill} = 0.2$ | 75.39 | 52.0 | 60.11 | 35.15 | 55.66 |
| $\lambda_{cross} = 0.05$ | 74.79 | 53.71 | 61.56 | 37.2 | 56.82 |
| $\lambda_{cross} = 0.1$ | **76.17** | **54.0** | 59.51 | 37.82 | **56.88** |
| $\lambda_{cross} = 0.15$ | 73.47 | 53.9 | 60.58 | 37.08 | 56.26 |

### A.4.3 SENSITIVITY TO STYLE DATASETS

To evaluate the sensitivity of our framework to the source of stylization, we conducted experiments where the style images for AdaIN were sampled from different domains within the PACS dataset. The out-of-domain generalization performance is reported in Table 10. Note that we intentionally exclude the "Sketch" domain as a style source. As discussed previously (Appendix A.4.1), its sparse, line-based nature lacks the rich textural and color statistics required for effective stylization via AdaIN, which could introduce confounding variables into our sensitivity analysis.

**Analysis.** Our results, presented in Table 10, reveal an insightful dependency on the nature of the style source domain. Using the "Art" domain, which is characterized by diverse and rich textural patterns, as the style source yields the strongest OOD performance. This confirms that our framework is robust when provided with a sufficiently expressive stylization signal.

Conversely, using the "Cartoon" or "Photo" domains leads to a noticeable degradation in performance. We hypothesize two primary reasons for this. First, for domains like "Cartoon" with large, uniform color blocks, the feature statistics captured by AdaIN are less expressive, leading to a less effective stylization and thus a weaker training signal. Second, for the "Photo" domain, the stylistic difference between the source content image and a photo-stylized version is minimal. This makes the self-distillation task too simple, reducing the incentive for the model to learn a sharp and robust disentanglement of content and style.

This finding highlights a key insight: the effectiveness of our pre-training principle relies not just on style variation, but on variation that is sufficiently rich and distinct from the content source to drive robust disentanglement. This suggests that curating a diverse and stylistically expressive style dataset is an important factor for optimal performance.

### A.5 ANALYSIS OF CONTENT SUBSPACE DIMENSIONALITY

To address the potential limitation of modeling content as a single direction, we extended our framework to support a multi-dimensional content subspace and investigated the effect of its dimensionality, $k$. Instead of using the average of student and teacher representations to define a single vector, we perform Singular Value Decomposition (SVD) on their cross-covariance matrix. The top-k left singular vectors are then used as an orthonormal basis for the k-dimensional content subspace. The content representation is subsequently computed by projecting the feature vector onto this subspace.

Table 9: Ablation studies of gradient flow from cross-reconstruction on PACS. Best methods are highlighted in **bold**, and all reported accuracies are in percent (%).

| | PACS | | | | |
| | Target Domain | | | | |
| **Method** | **photo** | **art** | **cartoon** | **sketch** | **avg.** |
|---|---|---|---|---|---|
| stop gradient | 73.77 | 51.46 | 58.3 | **39.78** | 55.83 |
| normal gradient | **76.17** | **54.0** | **59.51** | 37.82 | **56.88** |

Table 10: Out-of-domain accuracy on PACS when using different domains as the style source during pre-training. Best methods are highlighted in **bold**, and all reported accuracies are in percent (%).

| | PACS | | | | |
| | Target Domain | | | | |
| **Style Datasets** | **photo** | **art** | **cartoon** | **sketch** | **avg.** |
|---|---|---|---|---|---|
| photo | 62.51 | 45.94 | 56.19 | 36.96 | 50.40 |
| cartoon | 57.54 | 47.85 | 55.85 | **40.49** | 50.43 |
| art | **76.17** | **54.0** | **59.51** | 37.82 | **56.88** |

And the process of identifying the k-dimensional content subspace and projecting onto it is formalized as follows.

Let $F_A, F_B \in R^{B \times C}$ be the matrices of L2-normalized $[CLS]$ token representations for the teacher and student batches, respectively, where $B$ is the batch size and $C$ is the feature dimensionality. First, we compute the cross-covariance matrix between the two sets of representations:

$$\sum_{AB} = \frac{1}{B} F_A^T F_B$$

Next, we perform SVD on this matrix to find the principal directions of correlation:

$$U, S, V^T = SVD(\sum_{AB})$$

Where $U \in R^{C \times C}$ contains the left singular vectors as its columns.

The orthonormal basis for our k-dimensional content subspace, $v_c$, is formed by selecting the top-k left singular vectors from $U$:

$$v_c = U_{:,:k} \in R^{C \times k}$$

Finally, for any given feature vector $z \in R^C$, its content component $c \in R^C$ is computed by projecting $z$ onto this subspace:

$$c = v_c v_c^T z$$

The corresponding style component s is then derived as the orthogonal residual, $s = z - c$.

**Analysis.** According to the results on the PACS dataset are presented in Table 11, we found that increasing the dimensionality to $k = 2$ yielded a notable improvement over the single-vector baseline ($k = 1$). This suggests that a 2D subspace can capture a richer, more expressive representation of content than a single direction. However, further increasing the dimensionality to $k = 4$ resulted in a performance degradation, falling below even the $k = 1$ baseline. This finding indicates a "sweet spot" for the content subspace's capacity. While a single direction ($k = 1$) might be too restrictive, a higher-dimensional space ($k = 4$) may begin to capture redundant or even style-entangled features, effectively introducing noise into the content representation and harming the disentanglement process. This highlights the importance of choosing an optimal dimensionality and supports our choice of $k = 2$ as the best configuration for this task.

A.6 COMPUTATIONAL COST ANALYSIS

To quantify the computational overhead of our proposed framework, we compare its training efficiency with the standard MAE baseline. Both models were pre-trained on the PACS dataset (using

Table 11: Out-of-domain accuracy on PACS with varying content subspace dimensionality ($k$). Best methods are highlighted in **bold**, and all reported accuracies are in percent (%).

| | PACS | | | | |
| | Target Domain | | | | |
| Dimensionality k | photo | art | cartoon | sketch | avg. |
|---|---|---|---|---|---|
| $k = 1$ | 76.17 | 54.0 | **59.51** | 37.82 | 56.88 |
| $k = 2$ | **77.31** | **55.37** | 59.13 | **40.72** | **58.13** |
| $k = 4$ | 75.69 | 52.83 | 59.01 | 38.10 | 56.41 |

3 domains as source) for 1000 epochs on two NVIDIA RTX 4090 GPUs with a total batch size of 128. The results, calculated directly from the training logs, are summarized in Table 12.

Table 12: The comparison of computational cost, the primary overhead in HyGDL comes from the additional forward pass of the teacher encoder, cross-reconstruction and the SVD computation.

| Model | Hours | GPU-Hours | samples/sec | GB perGPU |
|---|---|---|---|---|
| MAE | 4.17 | 8.34 | 263.6 | 4.35 |
| HyGDL | 5.8 | 11.6 | 174 | 18.3 |

As shown in Table 12, our HyGDL framework introduces a notable computational overhead compared to the standard MAE. The total training time increases by approximately 40%, with a corresponding decrease in throughput. The peak GPU memory usage is also significantly higher. This overhead is an expected consequence of our training strategy, which involves multiple additional computations per step: a full forward pass through the teacher encoder for distillation, a second decoder pass for the cross-reconstruction objective, style-transfer based on AdaIN and the SVD computation for subspace identification.

It is crucial to note that this is a one-time pre-training cost. For all downstream applications and inference tasks, only the trained student encoder is utilized. Therefore, the inference cost of a HyGDL, in terms of latency, throughput, and memory footprint, is identical to that of a standard MAE model of the same architecture. We believe this investment in a more sophisticated pre-training process is a reasonable trade-off for the significant gains in out-of-domain generalization and the ability to learn robust, disentangled representations.

