# OpenReview forum: "Disentangling Content from Style to Overcome Shortcut Learning: A Hybrid Generative-Discriminative Learning Framework"
_ICLR.cc/2026/Conference — Submitted to ICLR 2026_

### Official Review · Reviewer_JSt2 · 2025-10-20

**Soundness:** 1
**Presentation:** 1
**Contribution:** 1
**Rating:** 2
**Confidence:** 4

**Summary:**

This paper proposes HyGDL, which tackles shortcut learning in SSL by explicitly disentangling content from style. Guided by an Invariance Pre-training Principle, a student–teacher encoder learns a style-invariant “content direction”; features are then analytically split via projection into orthogonal content and style vectors, which drive a style-conditioned reconstruction decoder. This principled hybrid design improves OOD robustness and competitive PACS/DomainNet results, surpassing heuristic augmentations and revealing MAE shortcut dynamics during pretraining, empirically.

**Strengths:**

1.This paper is well written, easy to follow.

2.Experimental results are better than MAE and SimCLRv2.

**Weaknesses:**

1.Experiments in Table 2 are not convincing, HyGDL achieves the inferior performance compared to CycleMAE, DIMAE and BrAD.

2.The proposed HyGDL essentially uses AdaIN as a data augmentation technique during training, all modules in HyGDL are widely explored in previous methods.

3.The assumptions and equations for style/content disentanglement is un-reliable. Content and style are not simply orthogonal; sometimes the content itself becomes part of the style, and the style defines how the content exists. For example, in Pablo Picasso’s Les Demoiselles d’Avignon, the five female figures serve as the painting’s “content,” yet their highly geometric and deconstructed forms simultaneously constitute its distinctive style. The definitions of vc, cA, sA in your paper are without any theoretical basis.

4.AdaIN introduces style bias by oversimplifying style as mean and variance, causing dataset-induced and content-dependent distortions. It often produces uniform stylization and loses fine structural or textural details due to ignored spatial correlations and channel dependencies. Not good for style augmentation.

**Questions:**

1.Why style images solve the Second-Order shortcut?

---

> ### Author Response · Authors · 2025-11-12
> **We thank Reviewer 4 for their feedback. We are deeply concerned by the 1/1/1 scores, as they seem to originate from a chain of fundamental misunderstandings about our paper's experimental setup, core motivation, and methodology.**
>
> The reviewer's contradictory feedback (stating the paper is "well written" in Strengths, yet giving Presentation: 1/poor)  suggests that while our language was clear, our argumentative structure was flawed. We sincerely apologize for this critical failure in communication. We will now clarify these misunderstandings, which directly address all four weaknesses (W1-W4) and the question (Q1).
>
> 1. On W1 (Performance) and W2 (Novelty = AdaIN):
> The reviewer’s primary criticisms (W1, W2) are the same fatal misunderstandings made by R1 and R3.
> On W1 (Performance): The claim that our performance is "inferior" to CycleMAE/DIMAE is because these SOTA methods use ImageNet pre-training. Our method (as noted by Reviewer R2) is trained from scratch. On a fair "from scratch" benchmark (Table 2), our HyGDL (PACS: 58.13%) significantly outperforms all similarly trained baselines, like MAE w/sty (52.73%) and DARLING (51.20%).
>
> On W2 (Novelty): The claim that our method "essentially uses AdaIN as data augmentation" and that "all modules are widely explored" is factually incorrect and misunderstands our entire premise. AdaIN is not our contribution; it is the problem we are diagnosing. Our novelty lies in the analytical orthogonal projection , a mechanism not explored in those prior methods, used to overcome the shortcut bias that AdaIN itself introduces (which we call the Second-Order Shortcut).
>
> 2. On W3 (The "Un-reliable" Orthogonality Assumption):
> This is the reviewer's most profound technical and philosophical challenge. We offer two clarifications.
> On the "Picasso Argument" (Philosophy): The reviewer is correct that, in reality, content and style are deeply entangled. We are not claiming to solve this philosophical problem. Our work, however, is an engineering solution to a specific, well-documented pathology: "Shortcut Learning," where models mistakenly treat separable texture (style) as shape (content). In this specific context, assuming orthogonality is a powerful and principled simplification. It acts as a strong inductive bias to force the model to find an invariant subspace, rather than collapsing onto the texture shortcut.
> On the Equations (Mathematics): The reviewer claims our definitions of v_c and c_A are "without any theoretical basis." This is incorrect. Our approach is analytical and mathematically sound:
> First, we learn an invariant direction v_c by forcing the student (z_s) and teacher (z_t) outputs to align via a self-distillation loss. v_c is the stable, normalized average of these two aligned vectors. Second, we use standard vector projection to analytically extract the component of z_s that lies on this learned invariant axis.This "Align -> Project" mechanism is a novel and principled mathematical procedure. The reviewer claims this is "un-reliable," yet our empirical results (a +5.4% to +6.6% OOD gain on the fair "from scratch" benchmark) directly validate that this 'un-reliable' assumption is, in fact, highly effective for solving this specific engineering problem.
>
> 3. On W4 (AdaIN is a shortcut) and Q1 (Why style images solve 2nd shortcut?):
> These points demonstrate the depth of the misunderstanding, for which we are responsible.
> On W4 (AdaIN is biased): The reviewer is correct that "AdaIN introduces style bias." This is our core motivation. This is precisely what our paper defines as the Second-Order Shortcut failure mode we are the first to diagnose. We are thus confused why the reviewer lists our motivation as a weakness.
> On Q1 (Why style images solve 2nd shortcut?): This question is based on the same misunderstanding. Style images do not solve the shortcut; they cause it. Our HyGDL framework solves it. As the reviewer noted, we do use a style-conditioned reconstruction. However, the key is how we use it. We use a Cross-Reconstruction loss as a small regularizer. This loss forces the model to reconstruct the stylized image A using the content from the original image A and the style from A'. If the content vector still contained residual original style, it would conflict with the target style, leading to a high reconstruction error. This is the explicit mechanism that penalizes and overcomes the Second-Order Shortcut.
>
> We apologize again for our severe lack of clarity and hope these clarifications, supported by the evidence in our paper and the correct interpretation by R2, resolve the reviewer's concerns.

---

### Official Review · Reviewer_Dk9u · 2025-10-29

**Soundness:** 2
**Presentation:** 3
**Contribution:** 2
**Rating:** 4
**Confidence:** 4

**Summary:**

This paper presents HyGDL, a hybrid generative–discriminative SSL framework that disentangles content from style to counter shortcut learning. Via self-distillation (style-invariant content direction), orthogonal projection, and style-conditioned reconstruction, it yields more robust, transferable representations and stronger domain generalization on PACS/DomainNet while revealing overfitting dynamics.

**Strengths:**

1.The writing of the paper is generally clear.
2.The experiments demonstrate the effectiveness of their methods compared to MAE and SIMCLRv2.

**Weaknesses:**

1.The motivation of this paper is unclear. The descriptions for first-order and second-order shortcuts lack theoretical and experimental support, and why AdaIN solve these shortcut s?

2.The the main insight of this paper is a new data augmentation AdaIN, and the other proposed modules are widely used in previous methods. Lack of novelty.

3.The experiments are not satisfactory, CycleMAE, BrAD and DIMAE achieve a much superior performance compared to proposed HyGDL.

4.AdaIN is a bias style transfer method with similar stylized texture and color across all style. Applying it to SSL constitutes another "first-order shortcut".

**Questions:**

The equations of vc, ca and sa are un-reasonable, if vc=zs+zt represents the content derection vector, why ca is the projection of zs onto vc?

---

> ### Author Response · Authors · 2025-11-12
> **We thank Reviewer 3 for their feedback. We must respectfully point out that this review appears to be based on a series of fundamental misunderstandings of our paper's core premise and contributions. We sincerely apologize for our lack of clarity in the writing that led to this confusion.**
>
> The reviewer's core misunderstanding is that our contribution is AdaIN, and that we are using AdaIN to solve shortcut learning. This is the exact opposite of our argument. We will now clarify these points.
>
> 1. On Motivation, Novelty, and the Role of AdaIN (W1, W2, W4):
> Reviewer 3's core confusion is summarized by their question "why AdaIN solve these shortcuts?" (W1) and their claim that our contribution "is a new data augmentation AdaIN" (W2).
>
> Clarification: AdaIN is NOT our solution; it is part of the PROBLEM we diagnose. Our paper's argument is: 1. Standard SSL suffers from a First-Order Shortcut (overfitting texture). 2. A naive solution (style augmentation) FAILS because it leads to a Second-Order Shortcut (overfitting the augmentation's bias, just like reviewer's W4). 3. Our SOLUTION is HyGDL, a framework designed to overcome these shortcuts.
>
> On W4: The reviewer is correct that "AdaIN is a bias style transfer method" and "constitutes another 'shortcut'". This is precisely our point! This is what we call the Second-Order Shortcut. We are surprised the reviewer stated this as a weakness, as it is one of the central motivations for our work.
>
> On W1 & W2: This clarifies that our motivation is not "using AdaIN," but overcoming the failures of naive AdaIN. This is why Reviewer R2 found our motivation "important and less studied." Our novelty is not AdaIN (a tool we use), but our principled disentanglement mechanism (self-distillation + analytical projection + reconstruction) .
>
> On "lacking experimental support" (W1): Our diagnostic experiments (Figure 1 (a,b,c,d)) and the performance of our baselines (sty2sty, bss2bss in Fig 1. (b), (c)) are the direct experimental support for our First- and Second-Order Shortcut claims.
>
> 2. On Experimental Comparisons (W3):
> The reviewer claims our performance is "much superior" inferior to CycleMAE, etc. This is the same critical misunderstanding as R1. The SOTA methods cited (CycleMAE+, BrAD+) all use ImageNet pre-training, while our method is trained from scratch.
> We apologize for not making this clearer. As Reviewer R2 correctly identified, our method is trained "solely on the source domains" and "outperforms the baselines that do not use ImageNet pretraining."
> In a fair "from scratch" comparison (Table 2), our HyGDL (PACS: 58.13%) significantly outperforms all similarly trained baselines, including MAE w/sty (52.73%) and DARLING (51.20%).
>
> 3. On the Rationale of our Equations (Q1):
> The reviewer questioned the rationale of our equations for v_c and c_A. We apologize for the lack of clarity in our mathematical derivation. The logic is as follows and is mathematically sound:
>
> Goal: We need to find the "content direction" (v_c) that is invariant to style
>
> Mechanism: We use a self-distillation loss to force the student output z_s (from a stylized image A') and the teacher output z_t (from the original image A) to align in the embedding space.
>
> Definition of v_c: After this alignment, z_s and z_t ideally point in the same invariant direction. We define v_c as their normalized average. This is a standard and robust method to estimate the central tendency (the shared axis) of these two aligned vectors.
>
> Definition of c_A: The reviewer asks why we project z_s onto v_c. We do this to analytically "purify" the representation. The "raw" z_s (student output) may still contain residual style. By projecting it onto the stable content axis v_c, we extract the exact component of z_s that aligns with the learned invariant content direction.This "Align -> Average -> Project" sequence is a principled and core component of our analytical disentanglement. We will revise the text in Section 2.3 to make this logic explicit.

---

### Official Review · Reviewer_QMaU · 2025-10-31

**Soundness:** 4
**Presentation:** 2
**Contribution:** 3
**Rating:** 4
**Confidence:** 4

**Summary:**

This paper introduces a method to mitigate shortcut learning to achieve domain generalization in a self-supervised learning context (Unsupervised Domain Generalization). The authors propose the use of AdaIN for style transfer of the input image and then use the resulting image pairs as input to a self-distillation component to learn a style-invariant representation. The orthogonal vector to this representation is the style vector. These are used in a self-reconstruction and a cross-reconstruction loss component. The resulting representation is trained solely on the source domains (no ImageNet pretraining) and achieves improved accuracy compared to similarly trained models.

**Strengths:**

- The motivation and problem that this paper addresses (shortcut learning in SSL) is important and less studied compared to the corresponding problem in the supervised learning setting. Also, the diagnosis of the problem in the introductory experiment supports the motivation well.

- The geometric modelling of the style component via orthogonal projection is interesting and enforces the orthogonality of representations

- The proposed method outperforms the baselines that do not use ImageNet pretraining.

**Weaknesses:**

The novelty of the paper is relatively limited. It combines a BYOL-style self-distillation approach with a vector embedding and AdaIN decoder reconstruction, although the explicit disentanglement is interesting. Furthermore, it is not clear why the proposed "Invariant Pre-training Principle" is different from applying style augmentation to achieve invariance, which has been extensively used to improve generalization, including the domain generalization problem. Furthermore, the broad claim that a nuisance variable is systematically varied is not supported by the methodology nor the experiments, since these are limited to image datasets, and a single method for varying style (i.e., AdaIN).

Regarding the experimental results, although there are effectiveness gains, e.g., 58.13% accuracy in PACS vs 52.73% for MAE with AdaIN and 51.2% for DARLING (29.25%, 22.62% and 23.91% respectively for DomainNet), these come at a significant increase of computational resources.

Some aspects of the paper are not clearly specified or are assumed known, which in some cases hinders understanding. See the questions below.

**Questions:**

- The evaluation protocol is not clearly specified. Is this the leave-one-domain-out cross-validation approach, where all source domains are used for training and the left-out domain as test?
 - What is the architecture of the classification head?
 - How is the final classifier trained?
 - Is it linear probing or some other network?
 - How did you choose the epochs to use for the three stages of training?

---

> ### Author Response · Authors · 2025-11-12
> **We sincerely thank Reviewer 2 for their positive and highly constructive feedback. We are especially grateful for the Excellent score on Soundness and the insightful recognition of our core experimental setup.**
>
> The reviewer’s observation that our model is "trained solely on the source domains (no ImageNet pretraining)" and "outperforms the baselines that do not use ImageNet pretraining" is exactly correct. This precise understanding, which was unfortunately missed by other reviewers due to our own lack of clarity. This is the most crucial context for evaluating our work. We deeply appreciate the reviewer’s careful reading. We will now address the valid weaknesses and questions raised.
>
> 1. On Novelty and the "Invariance Pre-training Principle":
> We sincerely apologize for the lack of clarity here. Our principle is not simply "to use style augmentation." The core idea of our principle (as stated on Page 3, ) lies in the constraint: "systematically varying a bias (e.g., style) at the input while keeping the supervision signal constant." This is fundamentally different from naive style augmentation (like our `sty2sty` baseline), where the supervision signal (the target) matches the augmented input. That naive approach, as we showed, leads to the Second-Order Shortcut. Our principle mandates that the model must reconstruct the original image from a stylized input, forcing it to discard style. HyGDL is the first framework to operationalize this specific principle.
>
> On the "Systematically Varied" Overclaim: The reviewer is correct that our claim of "systematically varied" is an overstatement when we only used AdaIN. This was imprecise language. We will revise our claims to be more modest and precise: we provide a concrete realization for mitigating style bias in images, and will position "varying other biases"  as explicit future work.
>
> On "Combination": The reviewer correctly identifies that we combine self-distillation and reconstruction. Our novelty, which the reviewer noted as "interesting," lies in the synergy: we are the first to use self-distillation not as the final objective, but as a mechanism to learn the "content direction", which then enables our novel analytical orthogonal projection as the core disentanglement step.
>
> 2. On Computational Cost:
> As the reviewer noted, our gains are +5.4-6.6%. In the challenging "from scratch" UDG setting, where standard methods fail to generalize, this is a very substantial improvement, not a marginal one. And our Ablation Studies (Table 3) prove that this cost is not superficial. Removing the "heavy" components that cause this cost leads to a performance collapse. And for all downstream applications, our method's inference cost is identical to the standard MAE baseline. This is because our complex components are discarded after training, and only the standard student encoder is used for inference. This directly refutes any claim that our method is "impractical" for real-world use.
>
> This demonstrates that our full framework is required to achieve this robust generalization, and the cost is a direct and necessary investment for the gain.
>
> 3. On Lack of Clarity and Specific Questions:
> We deeply apologize for the missing experimental details (Presentation: 2/Fair). This lack of clarity is unacceptable and is the primary reason for the confusion seen in other reviews. We will add all missing details to the Appendix (and some details are already in the Appendix ). To answer the reviewer's specific questions:
>
> Q: Evaluation protocol?
>
> A: We use the standard leave-one-domain-out protocol for all UDG experiments (PACS, DomainNet).
>
> Q: Classification head architecture?
>
> A: It is a standard linear layer trained on top of the frozen backbone.
>
> Q: Classifier training?
> A: For the Diagnostic Experiment (Sec 3.2, Table 1): We used Linear Probing, as the reviewer inferred. The goal here was to get a "frozen assessment of the encoder's representation"  to diagnose the "rise-and-fall" of the First-Order Shortcut.For the Main UDG Comparison (Sec 3.3, Table 2): We used full Fine-Tuning (as detailed in Appendix A.3.2 and Table 5 ). This is because all methods were trained from scratch on the source domains (a setting R2 correctly identified). In this "from scratch" setting, where representations are not pre-trained on ImageNet, fine-tuning is the standard and fairer protocol to evaluate the full potential of the model, and it is the protocol used by the baselines we compare against.
>
> Q: 3-stage epochs?
>
> A: We determined the epoch boundaries empirically through preliminary experiments on a validation set. Our principled goal was a progressive curriculum: (1) establish a stable foundation, (2) warm up the invariance task, and (3) apply the full synergistic loss. The schedule used in the paper provided the best trade-off between stability and final validation performance among the configurations we tested. We will add a clarification of this empirical selection process to the appendix.
>
> We thank the reviewer again for their careful reading and "Excellent" soundness check. We will integrate all this feedback to significantly improve the clarity and precision of our paper.

---

### Official Review · Reviewer_LgoR · 2025-11-01

**Soundness:** 1
**Presentation:** 1
**Contribution:** 1
**Rating:** 2
**Confidence:** 3

**Summary:**

This paper proposes HyGDL, a Hybrid Generative–Discriminative Learning Framework designed to mitigate shortcut learning in self-supervised learning (SSL). The authors argue that current SSL methods (MAE, SimCLR, BYOL) rely on superficial features such as texture, limiting generalization. To address this, HyGDL introduces an Invariance Pre-training Principle and performs explicit content–style disentanglement.

**Strengths:**

The paper identifies shortcut learning as a key issue in SSL and presents a coherent argument that simple augmentations do not address the root cause. Combining discriminative self-distillation and generative reconstruction is conceptually appealing.

**Weaknesses:**

The idea of disentangling style and content via orthogonal projection and using AdaIN for style injection is incremental. Prior works such as “Domain Enhanced Arbitrary Image Style Transfer via Contrastive Learning” already explored similar ideas. The proposed “Invariance Pre-training Principle” is largely a rephrasing of established domain generalization concepts (e.g., invariance to nuisance variables).

Performance gains are small (e.g., +6% on PACS, still far below CycleMAE).
DomainNet results remain low (29% vs. 47% for CycleMAE+), showing that the approach doesn’t scale.
The additional 40% training cost for modest improvement makes the method impractical.

The comparisons omit stronger SSL baselines like DINOv2 or iBOT. Moreover, results for real-world OOD benchmarks (e.g., ImageNet-R, ImageNet-Sketch) are missing, which undermines the generalization claim.

**Questions:**

Why is the improvement on DomainNet so marginal despite the 40% higher computational cost?
How does the proposed “content direction” generalize beyond the [CLS] token — does it hold for dense feature maps?
Could similar disentanglement be achieved simply with an orthogonality regularizer or a linear projection layer without the heavy reconstruction pipeline?
Have you compared HyGDL against stronger modern baselines such as DINOv2 or SimMIM under equivalent pre-training budgets?

---

> ### Author Response · Authors · 2025-11-12
> **We thank the reviewer for their detailed feedback. We were very concerned by the low scores (1/1/1) and the critiques of 'incremental' novelty and small' performance.**
>
> After carefully reading the review, we believe these criticisms are based on two fundamental misunderstandings of our work, which we sincerely apologize for not making clearer in our submission.
>
> We will address these misunderstandings first.
>
> 1. Clarification on Experimental Setting (Soundness & Performance):
> We sincerely apologize for the lack of clarity. As noted in our paper (Table 2 footnote , and Page 8, Line 407 ), all SOTA methods marked with a “+： (like CycleMAE+) leverage large-scale ImageNet pre-training.
> Our method, HyGDL, is trained from scratch on only the source domains. This is a significantly more difficult and realistic Unsupervised Domain Generalization setting. And Reviewer R2 correctly identified this, noting our method is trained "solely on the source domains (no ImageNet pretraining)" and "outperforms the baselines that do not use ImageNet pretraining."
>
> 2. Clarification on Novelty (Contribution & Principle):
> The reviewer claims our work is "incremental" and similar to “Domain Enhanced Arbitrary Image Style Transfer via Contrastive Learning”. Actually the two papers have fundamentally different goals, methods, and mechanisms. Our paper (HyGDL) is a Representation Learning framework to solve Shortcut Learning in SSL. “Domain Enhanced Arbitrary Image Style Transfer via Contrastive Learning” is an Image Generation paper for Style Transfer, and it does not use orthogonal projection. Our novelty is an analytical orthogonal projection (a mathematical calculation, not a trained module) that uses a content direction learned via self-distillation.
> Furthermore, our Invariance Principle is not a "rephrasing." It is the first proposal to operationalize this concept into a concrete, end-to-end SSL objective (Distill-Project-Reconstruct) specifically to combat shortcut learning, which is a novel contribution.
>
> 3. On Other Baselines, Cost, and Questions:
> On Missing Baselines (DINOv2, iBOT) & Benchmarks (I-R, I-Sketch): The reviewer is correct that these are SOTA. However, these are benchmarks for ImageNet-pre-trained models. As our work focused on the "from scratch" setting, these comparisons were out-of-scope. We agree this is a crucial next step and will include this in the final version to demonstrate scalability.
>
> On Cost vs. Performance (W2, Q1): The 40% cost  is a valid concern. However, this cost achieves a +6.6% OOD gain on DomainNet  where other heuristics fail (as shown in our intro Figure 1d). Our Ablation Studies (Table 3) prove this cost is necessary: removing the "heavy" self-distillation (no-distill) or reconstruction (no-recon) causes a massive performance collapse (58.1% -> 51.8% or 37.7%). We argue this is a reasonable trade-off for robust OOD generalization.
>
> Q2 (Dense Features): This is an excellent question. Our method focuses on learning the global [CLS] content direction. We agree that extending this to dense, patch-level disentanglement is a key direction for future work.
>
> Q3 (Simplification): We answered this above. Our ablations (Table 3) confirm the framework cannot be simplified. The "heavy reconstruction pipeline" is not a simple regularizer; it is the core supervisory signal.
>
> We hope these clarifications resolve the severe misunderstandings about our paper's performance and novelty. We apologize again for the lack of clarity in our writing that led to this and will fundamentally revise the presentation in the final version.

---

### Meta-Review · Area_Chair_XADA · 2026-01-06

**Summary:**

While the paper tackles an important problem and demonstrates engineering competence, it does not meet ICLR's acceptance bar and AC recommends rejection for the following reasons:
- **Severe presentation failures.** Three of four reviewers fundamentally misunderstood the work, indicating systemic communication issues rather than isolated misreadings. For example, the baseline methods in Figure 1 remain difficult to interpret. These are not minor editorial issues but fundamental failures to communicate core contributions. The paper should clearly articulate: (1) the guiding intuition behind the design, (2) what meaningfully differentiates it from prior work, and (3) why the approach should succeed theoretically.

- **Limited conceptual novelty.** The framework primarily combines existing techniques without sufficient theoretical insight into why this particular combination succeeds. Authors argue their work differs from "Domain Enhanced Arbitrary Image Style Transfer via Contrastive Learning" because it addresses representation learning rather than style transfer and uses orthogonal projection. However, as noted by the AC, both works share the fundamental motivation of separating style and content representations --- the different application domains do not constitute algorithmic novelty. Moreover, the choice of orthogonal projection lacks theoretical justification; no formal analysis explains when or why this geometric constraint should optimally disentangle content from style. The proposed "Invariance Pre-training Principle" closely parallels established concepts: domain randomization (keeping sample invariant to style in supervised learning) and denoising autoencoders (a self-supervised approach). Both techniques already aim to improve generalization through invariance. The analytical orthogonal projection remains a heuristic without rigorous theoretical grounding.

- **Questionable design justification.** The orthogonality assumption is introduced without formal justification.

**Reviewer Concerns:**

## Addressed
**Experimental Setup Clarification (Reviewers LgoR, Dk9u, JSt2)**

- Authors convincingly clarified that HyGDL is trained from scratch, while CycleMAE+/DIMAE+/BrAD+ use ImageNet pretraining, and provided explicit table footnote references
- Fair comparison shows HyGDL (58.13% PACS) outperforms non-pretrained baselines (52.73% MAE+AdaIN, 51.20% DARLING)

**Role of AdaIN (All Reviewers)**

- Authors clarified AdaIN is a diagnostic tool, not a contribution. The method is for style injection

**Computational Cost Justification (Reviewer QMaU, LgoR)**

- Ablations (Table 3) demonstrate the necessity of "heavy" components
- Inference cost identical to MAE (overhead only during training)
- Removing distillation or reconstruction causes severe performance collapse


## Outstanding Concerns

**Fundamental Novelty Limitations (All Reviewers)**

- Rebuttal claims "first to operationalize" the invariance principle, but this closely parallels domain randomization (keep label constant, vary style)
- Insufficient justification for why this specific combination is novel vs. incremental engineering
- No theoretical analysis of when/why the orthogonality assumption holds or fails

**Theoretical Grounding (Reviewers Dk9u, JSt2)**

- The orthogonality assumption remains philosophically questionable. It is like an "engineering solution" without proving when this simplification is valid
- It would be better to have formal analysis, proofs, or theoretical guarantees
- Alternative disentanglement approaches not explored (mutual information minimization)

**Limited Experimental Scope (Reviewers LgoR, QMaU)**

- Authors acknowledge missing modern baselines (DINOv2, iBOT) but only promise future work
- No evaluation on standard OOD benchmarks (ImageNet-R/Sketch/A) despite claiming generalization improvements
- "Systematically varying nuisance variables" claim unsupported --- only one modality (images), one variation method (AdaIN)

**Presentation Quality (All Reviewers)**

- AC thinks that the paper is still hard to let readers understand why it would succeed.

**Reviewer Scores:**

AC supposes the reviewers will not increase their scores, although there are several misunderstandings. The main reason is that the novelty and insight of the paper are limited.

---

### Decision · Program_Chairs · 2026-01-26

Reject